# Functional Decline in the Cancer Patient: A Review

**DOI:** 10.3390/cancers14061368

**Published:** 2022-03-08

**Authors:** Jaidyn Muhandiramge, Suzanne G. Orchard, Erica T. Warner, Gijsberta J. van Londen, John R. Zalcberg

**Affiliations:** 1Department of Epidemiology and Preventive Medicine, School of Public Health and Preventive Medicine, Monash University, Melbourne, VIC 3004, Australia; jaidyn.muhandiramge@monash.edu (J.M.); suzanne.orchard@monash.edu (S.G.O.); 2Clinical and Translational Epidemiology Unit, MGH Cancer Center, Massachusetts General Hospital, Harvard Medical School, Boston, MA 02114, USA; ewarner@mgh.harvard.edu; 3Department of Medicine, University of Pittsburgh, Pittsburgh, PA 15260, USA; vanlondenj@upmc.edu; 4Department of Medical Oncology, Alfred Hospital, Melbourne, VIC 3004, Australia

**Keywords:** cancer, elderly, functional decline, functional status, frailty, ageing

## Abstract

**Simple Summary:**

Older patients with cancer are more likely to experience a deterioration in their functional status than are their similar-aged, cancer-free counterparts. Such functional decline can be accelerated by cancer treatment. With adequate long-term care comprising routine functional assessment and evidence-based interventions, functional status is likely to be maintained, or the rate of decline slowed. Mitigating the risk of functional decline is vital given its impact on quality of life and mortality.

**Abstract:**

A decline in functional status, an individual’s ability to perform the normal activities required to maintain adequate health and meet basic needs, is part of normal ageing. Functional decline, however, appears to be accelerated in older patients with cancer. Such decline can occur as a result of a cancer itself, cancer treatment-related factors, or a combination of the two. The accelerated decline in function seen in older patients with cancer can be slowed, or even partly mitigated through routine assessments of functional status and timely interventions where appropriate. This is particularly important given the link between functional decline and impaired quality of life, increased mortality, comorbidity burden, and carer dependency. However, a routine assessment of and the use of interventions for functional decline do not typically feature in the long-term care of cancer survivors. This review outlines the link between cancer and subsequent functional decline, as well as potential underlying mechanisms, the tools that can be used to assess functional status, and strategies for its prevention and management in older patients with cancer.

## 1. Introduction

Functional status captures an individual’s ability to perform the normal activities required to maintain adequate health and meet basic needs [1]. A decline in functional status is an inevitable part of normal ageing, with the rate of functional decline steadily increasing over a person’s lifespan [2]. This decline, however, is accelerated in individuals with cancer, with this cohort experiencing rapid deterioration following a cancer diagnosis and treatment, compared to their cancer-free counterparts [3]. Not only does functional decline impact quality of life, impairment in functional status can also lead to increased comorbidity burden, greater dependency on carers, institutionalisation, and increased mortality [4,5,6].

With early detection and the use of evidence-based interventions, the risk of functional decline in cancer survivors can be at least partly mitigated. Despite this, assessment of functional status rarely plays a role in the routine clinical care of older patients with cancer, particularly if they are not actively receiving treatment. With an ever-growing population of cancer survivors due to advances in anti-cancer treatment [3], it is vital that functional decline is addressed to ensure that patients with cancer are provided the opportunity to maintain a high quality of life and to live independently.

Functional decline in the older cancer patient is a broad and complex topic that features interplay between several key factors, including age, cognitive function, comorbidities, symptom burden, and medications. This narrative review therefore aims to explore the current evidence linking cancer and functional status and provide an overview of its assessment, the mechanisms driving it, and strategies to prevent and manage functional decline in older patients with cancer. The search strategy for this review can be found in the Appendix A: Literature search strategy.

## 2. Assessment of Functional Decline

Functional decline is typically assessed through a direct measure of an individual’s capacity to independently fulfil their activities of daily living (ADLs) [1]. ADLs are typically categorised as either basic (tasks required for normal day-to-day functioning), or instrumental (tasks that are not necessarily essential but allow for one to live independently). Basic ADLs include tasks such as ambulation, eating, dressing, toileting, and personal hygiene. Instrumental ADLs are typically more complex and include managing finances, shopping, transportation, home maintenance, communication, and managing medications [7].

It should be noted that frailty, typically defined as a “state of reduced physiologic reserve”, is not synonymous with functional decline, although the two frequently measure domains which overlap [8,9,10]. Frailty is a broader concept than functional decline and measures vulnerability to functional decline as opposed to functional decline itself—its clinical utility therefore lies in its ability to predict susceptibility to disability [11]. While functional decline can be observed through the assessment of ADLs, frailty is often initially clinically silent. Its development can instead be identified through markers such as nutritional status, physical activity, mobility, energy, strength, cognition, mood, and social support. In this regard, the assessment of frailty requires a more comprehensive ‘geriatric assessment’ beyond the assessment of ADLs [8]. The Fried frailty phenotype is typically considered to be the gold-standard definition for the measurement of frailty and defines it as the presence of three or more of the following: unintentional weight loss (at least 10 pounds/4.5 kg in the past year), weakness (grip strength in the lowest quintile adjusted for sex and body mass index), slowness (walking time in the lowest quintile adjusted for sex and height), a low level of physical activity, and self-reported exhaustion [12]. Frailty is certainly seen at greater rates in patients with cancer [13,14,15]. While this review will focus on the more clinically apparent functional decline, many of the measures used to assess, mitigate, and manage functional decline can also be applied to a frailty outcome.

There are several measures of functional status that can be used in older patients with cancer; these tools are summarised in Table 1. These include instruments such as the Barthel index, the Eastern Cooperative Oncology Group performance status (ECOG), the Katz index of independence in activities of daily living scale (ADL), the Instrumental Activities of Daily Living Scale (IADL), the Rosow–Breslau health scale, and the Karnofsky performance status scale (KPS) [8,9,10]. Performance status, measured by an assessment of ECOG scores (0–4) is particularly relevant in oncology and is often used to measure suitability for anti-cancer treatment [16]. Physical performance measures such as the Timed Up and Go test (TUG), grip strength, 6-min walk test (6MWT), and gait speed can be used as surrogate markers of functional status as physical strength is often necessary to perform ADLs (e.g., adequate grip strength is required for personal care) although this is not always the case, given that some level of function can be maintained in the presence of poor physical performance [8,9,10,17]. The benefit of physical performance measures, however, lies in their measurement requiring direct observation rather than self-report, providing a more objective measure of a patient’s function.

The comprehensive geriatric assessment (CGA) is a more holistic alternative to the above tools that assesses several domains, including medical conditions, medications, nutritional assessment, cognitive status, mental health, social circumstances, environment, and functional status. Given the variety of domains it assesses, the CGA provides a broader overview of older patients with cancer and captures their susceptibility to further decline, as opposed to solely assessing functional status. The CGA bears particular clinical relevance having been recommended by the American Society of Clinical Oncology (ASCO) for assessing frailty in patients over 65 years of age receiving chemotherapy [8,18]. Other tools that provide a broader assessment of older patients and have been validated in cancer cohorts include the 36-item short form survey (SF-36; a quality-of-life assessment) [19], Geriatric 8 (G8) [20], Vulnerable Elders Survey-13 (VES-13) [20], Fried frailty criteria [21,22], Senior Adult Oncology Program 2 tool [23], Groningen Frailty Indicator (GFI) [21], and Rockwood Clinical Frailty Scale [21]. Of these, Garcia et al. recommend the G8 for screening older patients with cancer, given a high level of evidence supporting its sensitivity and specificity when screening for vulnerabilities [20] although it is noted that this tool is a screening measure that provides a less comprehensive view of a patient’s holistic health and is instead meant to identify patients who require further assessment using a more detailed tool. The authors recommended the VES-13 as an effective screening alternative in resource-poor settings [20]. The Cancer and Aging Research Group (CARG) also provides an online assessment tool for clinicians that combines the KPS scale, TUG, and Blessed Orientation–Memory–Concentration test (a cognitive function assessment aimed at assessing the contribution of cognitive decline on functional ability) [24]. These geriatric assessments tend to provide not only an indication of declines in ADLs, but also frailty and the susceptibility of a patient to future disability.

**Table 1 cancers-14-01368-t001:** Tools that can be used to assess functional status in older patients with cancer.

Instrument	Method of Administration	Domains Assessed	Comments
Functional status			
Barthel Index [25] (Basic ADLs)	Patient-reported or direct observation	Feeding, toileting, bathing, dressing, and undressing, toilet transfers, incontinence, bed transfers, and ambulation	Intended for patients with stroke, neuromuscular disorders, musculoskeletal disorders, and cancer.
Eastern Cooperative Oncology Group Performance Status (ECOG) [26]	Patient-reported	Percentage of day spent ambulatory or in bed	5-point scale, where 0 is “Fully active” and independent and 5 is “Dead”. Commonly used in oncology due to its simplicity [27]. Tends to have minimal direct input from the patient. Noted by the International Society of Geriatric Oncology (SIOG) to be a poor marker of function as functional impairment can occur in the presence of good performance status [28].
Karnofsky Performance Status Scale (KPS) (Both instrumental and basic ADLs)	Patient-reported	Activity, work, self-care	10–100-point scale, gold-standard measurement of performance status in cancer. Thorne-modified KPS better suited to community-based and palliative care settings [29], while Australia-modified KPS is better suited to settings with multiple venues of care across both inpatient and outpatient settings [30]. Noted by SIOG to be a poor marker of function as functional impairment can occur in the presence of good performance status [28].
Katz Index of Independence in Activities of Daily Living Scale (ADL) [31] (Basic ADLs)	Patient-reported	Bathing, dressing, toileting, transferring, continence, and feeding	Most commonly used instrument in studies assessing activities of daily living in adults with cancer [32]. Shortened versions are often used due to length: modified Katz-1 assesses dressing, bathing, transferring, eating, and toileting, but does not assess continence; modified Katz-2 assesses the original six domains in the Katz ADL scale, as well as walking across a small room [27].
Lawton Instrumental Activities of Daily Living Scale (IADL) [33]	Patient-reported	Ability to use telephone, shopping, food preparation, housekeeping, laundry, transport, responsibility for medications, and finances	Second-most commonly used instrument used in studies assessing activities of daily living in adults with cancer [32].
Rosow–Breslau Health Scale [33]	Patient-reported	Ability to do heavy housework, walk up and down stairs, and walk half a mile	Simple 3-point scale that can be easily implemented in the clinical setting. Less commonly used in patients with cancer and in oncology research.
Functional Independence Measure (FIM) [34]	Direct observation	Self-care, sphincter control, transfers, locomotion, communication, and social cognition	Used for evaluation in the rehabilitation of patients post-stroke, traumatic brain injury, spinal cord injury, or cancer.
Frail Elderly Functional Assessment Questionnaire (FEFA) [35]	Patient-reported	Mobility, transfers, housework, meal preparation, finances, telephone use, eating, dressing, personal hygiene, and medication management	Older, less-widely used tool. Validated against Katz ADL, IADL, and Barthel Index [36].
Elderly Functional Index (ELFI) [37]	Patient-reported	Physical functioning, role functioning, social functioning, and mobility	Newer tool derived from functional domains of common quality of life instrument European Organisation for Research and Treatment (EORTC) Quality of Life Questionnaire Core-30 (QLQ-C30). Suggested for use as an endpoint of functional status in clinical trials or in clinical practice.
Physical performance measures			
Grip strength	Direct observation	Forearm strength	Requires a dynamometer for testing. Poorer scores are associated with poorer health-related quality of life [38] and increased mortality [39] in patients with cancer.
Gait speed [40]	Direct observation	Walking speed over a short distance, typically 4, 6, 8, or 10 m	Poorer scores are associated with decreased survival outcomes and treatment-related complications in cancer survivors [41]. Requires a stopwatch, although electronic gait mats or automatic timing devices provide more accurate assessments [40].
6-Minute Walk Test (6MWT) [17]	Direct observation	Aerobic capacity and endurance over six minutes of walking	Good measure of cardiorespiratory fitness. Validated for use in patients with cancer [42]. Does not require specialised equipment, but does require a stopwatch and a walkway of known length.
Timed Up and Go Test (TUG) [43]	Direct observation	Gait speed and mobility: measures the time taken to rise from a chair, walk three meters, turn around, walk back to the chair, and sit down while turning 180 degrees	Poorer scores are associated with decreased survival outcomes, treatment-related complications, and functional decline in cancer survivors [41]. Can be used as a substitute measure for gait speed. Does not require specialised equipment.
Short Physical Performance Battery (SPPB) [44]	Direct observation	Lower limb muscle strength, balance, and mobility	Poorer scores are associated with decreased survival outcomes, treatment-related complications, and functional decline in cancer survivors [41]. Can be used as a substitute measure for gait speed. Does not require specialised equipment.
Physical Performance Test (PPT) [45]	Direct observation	Writing, eating, dressing, grip strength, mobility, dexterity, communication, upper limb function, and balance	Requires various household items for assessment. Direct comparison with the KPS scale indicates that the PPT is more accurate in measuring functional status in older patients with cancer [46].

## 3. Impact of Cancer on Functional Decline

There is substantial evidence linking cancer to reduced ADLs in older adults although these studies tend to be small, given that most cancer clinical trials focus on disease-related outcomes, such as overall survival or progression-free survival, rather than ageing outcomes such as functional decline. Notably, fewer studies assess a longitudinal decline in functional status, likely due to the logistical difficulty of capturing change in functional status over time. Those that do typically have a focus on post-treatment functional decline, with few studies adjusting for the impact of anti-cancer treatment and thereby assessing the influence of the tumour itself on functional status. Additionally, most studies capture functional declines that occur during the acute phase of a cancer, during active treatment, or immediately post-treatment. Conversely, there is a paucity of literature investigating ongoing impairment of function in cancer survivors well after curative treatment has taken place. Nonetheless, some of the literature on this topic is outlined below, with a preference for landmark or recent studies with large sample sizes or systematic reviews, that adjust for anti-cancer treatment, and involve comparison with cancer-free controls.

### 3.1. Prevalence of Functional Impairment in Older Patients with Cancer

Neo et al. provides a general overview of ADLs in their systematic review and meta-analysis of 43 studies comprising 19,246 older adults with cancer. They reported that the mean prevalence of impaired ADLs was 36.7% in the overall cohort, with the most commonly affected basic ADLs being personal hygiene, ambulation, and transfers. They further observed that the most commonly affected instrumental ADLs were housework, shopping, and transportation. While the studies within the meta-analysis do not make comparisons with cancer-free controls, track changes in function over time, or address whether impairment of ADLs occurred as a result of the cancer and/or cancer treatment, it nonetheless offers a broad picture of disability in the older patient with cancer [32]. Blackwood et al. provides a similar snapshot of functional status in older cancer survivors using the surveillance, epidemiology and end results (SEER) national cancer registry and Medicare Health Outcomes Survey. In those older than 85 years of age, breast and prostate cancers conferred the greatest risk of impaired functional status. Functional impairment typically increased proportionally to stage in breast, colorectal, lung, and prostate cancers. As with the previous study, this analysis did not compare the cancer cohort to a control group; however, it did provide an overview of the varying impact of cancer type and stage on functional status [47].

### 3.2. Cancer-Related Functional Decline

Several studies do, however, demonstrate longitudinal declines in functional status in older adults. Reeve et al. demonstrated a greater decline in physical function in patients with prostate, breast, bladder, colorectal, kidney, and lung cancers, with the latter showing the greatest deterioration although the study did not adjust for treatment variables [48]. Using a modified Rosow–Breslau questionnaire, Petrick et al. found functional declines in patients with lung, prostate, breast, and colorectal cancer within one year of diagnosis when compared to a cancer-free group. These deficits had not returned to baseline levels after one year in the groups with either lung or colorectal cancer, a finding the authors attribute to either early death due to increased mortality in these cancer types or disease-related declines in physical function [10]. Using the ADL and IADL scales, van Abbema et al. similarly found a cancer diagnosis to be a significant predictor of functional decline. Nearly half (43.6%) of the elderly cancer group showed functional status declines compared to 28.1% of the elderly non-cancer group [49]. These findings have been replicated in patients with lymphoma; La Carpia et al. reported statistically significantly poorer functional status scores in cancer survivors when compared to cancer-free controls [50].

Functional decline in patients with lung cancer is a particularly well-documented phenomenon [51,52]. Granger et al. demonstrated poorer scores in various functional outcomes in patients with non-small cell lung cancer when compared to age-matched cancer-free controls, with cancer being associated with poor 6MWT scores (84% of predicted distance) and quadriceps strength (mean difference 4.8 kg, 95% CI 1.6–8.1) at baseline. The study cohort also performed poorly on the functional components of quality-of-life measurements including the SF-36 and European Organisation for Research and Treatment (EORTC) quality of life questionnaire. Moreover, these patients experienced a regression in self-reported physical activity, the 6MWT (84% of predicted distance to 69%, *p* = 0.02), quadriceps strength (−3.9 kg, 95% CI −5.2, −2.6), and grip strength (−2.7 kg, 95% CI −4.6, −1.4) over the course of six months. The patients with lung cancer also demonstrated below-average baseline results in the 6MWT test and grip strength, suggesting an intrinsic impact of cancer as opposed to cancer treatment [51]. Decoster et al. similarly reported decreases in both ADL and IADL scores in nearly half of a cohort of older patients with newly diagnosed lung cancer after 3 months follow-up [52]. This phenomenon may be at least in part linked to patient physical loss of lung capacity impacting exercise tolerance and by extension, functional capacity. The impact of thoracic radiation is likely to compound this given the link between such radiation and cardiorespiratory function [53,54].

### 3.3. Cancer Treatment-Related Functional Decline

#### 3.3.1. Systemic Therapy

The impact of several modalities of cancer treatment on functional decline is similarly evident. In older patients receiving first-line chemotherapy, 16.7% experienced functional decline as measured via a comprehensive geriatric assessment pre- and post-treatment [55]. Hurria et al. demonstrated declines in physical function post-adjuvant chemotherapy in older patients with breast cancer. While nearly half had recovered by 12 months, almost one-third had ongoing decline after this period. Factors associated with resilience to functional decline after 12 months included strong social support and lower nodal burden, while baseline dyspnoea and poor appetite predicted persistent decline. The authors also suggest that early interventions aimed at improving functional status may play a role in a patient’s ability to ‘bounce back’ [56]. Kenis et al. similarly reported functional decline in nearly one-third of older patients receiving chemotherapy for various cancer types [57]. Similar declines can be seen in patients receiving hormonal therapy. Alibhai et al. demonstrated declines in grip strength in patients receiving androgen deprivation therapy for prostate cancer compared to cancer-free controls in a longitudinal assessment with testing at baseline, 3, 6, and 12 months. They also noted decreases in the physical function component of the SF-36 over the course of the study in the cancer group, while they found increases in the control group [58].

#### 3.3.2. Radiotherapy

Radiotherapy appears to have a similar impact on functional status. In their analysis of patients with lung cancer, Decoster et al. reported that radiotherapy was a statistically significant predictor of decline [52]. Ursem et al. corroborated these findings in an analysis of older patients with prostate cancer, demonstrating a decrease in minimum data set ADL score from the beginning of radiation to 3 months after, and then again from 3 months to 6 months post-treatment [59]. Notably, there is a paucity of studies stratifying functional decline outcomes by radiation location or dose. Given the impact of thoracic radiation on cardiorespiratory function [53,54], it may be that functional decline post-radiotherapy varies depending on where radiation is delivered.

#### 3.3.3. Surgery

The impact of surgery on ADL disability post-surgery is less clear. Amemiya et al. reported a transient decrease in functional status at 1-month post-operation for oesophageal or colorectal cancer, with recovery of nearly all patients by 6 months post-operation [60]. van Egmond et al. reports similar findings in a group of oesophageal cancer survivors post-oesophagectomy, with a high number of postoperative complications, but an overall return of functional status to baseline after 3 months [61]. Conversely, Tang et al. demonstrated a functional decline rate of 56–60% amongst 1-year breast cancer survivors post-surgery although this cohort comprised nursing home residents who were likely to have a poorer baseline functional status than the general population [62]. In a similar cohort of nursing home residents, nearly one-quarter of patients had persistent functional decline 1-year post-colectomy for colorectal cancer [63]. Given that patients typically require a reasonable functional baseline to be considered fit for surgery, patients receiving anti-cancer surgery may be inherently less likely to experience further decline post-operatively.

## 4. Mechanisms Driving Functional Decline

The development of functional decline in patients with cancer is likely multifactorial, with shared risk factors, social factors, comorbidities, tumour-related factors, and treatment all playing a role. In older adults, functional decline is likely to already be occurring as a normal consequence of ageing, irrespective of a cancer diagnosis, with the rate of functional decline generally steepening with increasing age [2]. Buchner et al. describes an accelerated ageing model that can be easily applied to the functional decline seen in older adults with cancer (Figure 1). Here, patients slowly lose function as they age, with poor lifestyle behaviours and acute insults, such as cancer and anti-cancer treatment, accelerating this process. This decline can be tolerated while the patient has ‘physiologic reserve’, the capability of an individual to tolerate stressors, until they reach a point at which functional disability occurs. This model recognises the slow decline of function over the lifespan and acknowledges that acute insults, such as cancer, are not the sole drivers of functional impairment in older patients with cancer, but instead, accelerate a pre-existing decline.

While some of these represent cancer- and treatment-related factors, the role of shared risk factors in functional decline in older patients with cancer must be acknowledged. For example, smoking is a strong risk factor for lung function decline [64] and has an adverse impact on functional status irrespective of cancer status [65]. Given that smoking is strongly associated with a number of cancer types, with nearly 80% of lung cancers being caused by smoking [66], older patients with cancer with a smoking history may have experienced smoking-related functional decline, regardless of their cancer diagnosis or treatment. These patients may also have concomitant smoking-related lung disease, such as chronic obstructive pulmonary disease, a condition independently linked with poorer functional status [67]. Obesity, another significant risk factor for cancer, particularly in older, post-menopausal women [68], can independently accelerate functional decline [69].

Some of the predictors of functional decline in patients with cancer, both cancer- and non-cancer related, are described in Table 2. Note that each factor may not predict functional decline across all cancer types and demographics although many are relevant to the general older cancer-survivor population.

Several tumour-related factors can contribute to functional decline although this varies between cancer types. In patients with primary lung cancer or multiple lung metastases, for example, the replacement of lung volume and subsequent reduction in pulmonary function can adversely impact functional status [72]. Patients with primary or secondary brain malignancies can experience disability in function due to motor or sensory deficits [73]. Similarly, patients with metastatic spinal cord compression are often afflicted by functional deficits [74]. In cancers not impeding on organ structures, tumour-related symptoms such as pain, fatigue, and depressive symptoms are likely to be the primary driving mechanism in functional impairment [75]. This may be particularly common in haematological malignancies, considering the prevalence of anaemia and its subsequent impact on a patient’s energy levels [76]. Given that as many as 38% of patients with cancer report moderate-to-severe pain, with fatigue and depressive symptoms being similarly common, recognising the impact of such symptoms plays an important role in any comprehensive assessment of functional status [77].

Cancer treatment is likely to be the main contributing factor to functional decline in patients with cancer, with exact mechanism varying between treatment modalities. Across all treatment types, fatigue is a common symptom that can result in functional deficits; prevalence estimates of fatigue during treatment can be anywhere from 25% to 99%, with up to one-third experiencing fatigue for as many as 10 years post-cancer diagnosis [78]. Chemotherapy-related toxicity is also incredibly common. Common toxicities, such as nausea and vomiting, diarrhoea, anaemia secondary to myelosuppression, peripheral neuropathy, vestibular dysfunction, weakness, and fatigue, are all likely to be drivers of functional impairment [79,80,81]. Hormonal therapy can have similarly debilitating side effects: In patients receiving androgen deprivation therapy for prostate cancer, for example, weakness and muscle wasting is common [82]. More targeted treatment modalities, such as radiotherapy, are likely to have less impact on functional status than their systemic counterparts. The nature of radiotherapy toxicity is both site- and dose-dependent but can result in complications such as cardiac toxicity limiting exercise tolerance, mucositis impacting personal care and eating, and fatigue that can have a pervasive impact on ADLs, although the latter two tend to be acute and resolve shortly after treatment [83]. Similarly, surgery is less likely to cause persistent functional impairment in older patients with cancer, with most patients returning to baseline functional status within months after an operation [60]. However, patients with post-operative complications and a prolonged length of hospital stay can suffer from accelerated bone loss, malnutrition, cognitive decline, and deconditioning, all factors that can independently contribute to declines in function post-discharge [84].

## 5. Clinical Implications

One of the most significant clinical implications of functional decline in older patients with cancer is its impact on health-related quality of life (HRQOL), perhaps best evidenced by the fact that one of the most commonly used tools for measuring HRQOL in patients with cancer, the EORTC Quality of Life Questionnaire–Core 30 (QLQ–C30), features a physical function section [85]. In a cohort of older breast cancer survivors, Mogal et al. found that HRQOL outcomes were most dependent on impairment in ADLs, as opposed to socioeconomic status or cancer-related factors [86]. Reeve et al. similarly found post-cancer deterioration across multiple domains of HRQOL using the SF-36 tool, with lung cancer demonstrating the greatest decline relative to a cancer-free control group [48]. In Deschler et al.’s analysis of 200 older patients with cancer post-oncologic surgery, ADL and IADL scores were correlated with global quality of life [87]. Borggreven et al. similarly reported deficits in HRQOL and functional status even prior to the commencement of treatment [88]. Anti-cancer treatment, including chemoradiotherapy [89] and surgery [87], appears to have an additional impact on independence-related quality of life. Given that the majority of cancer patients place at least equal weight on quality of life versus length of life, the impact of functional decline on quality of life is an important consideration [90].

Mortality is a similarly important consideration when examining the clinical implications of functional decline, with many measures of functional status carrying associations with increased mortality across several cancer types. Braun et al.’s study of patients with non-small cell lung cancer using the EORTC QLQ-C30 tool established that several of the survey’s domains, including physical functioning, were statistically significant predictors for survival. It also found a statistically significant 9% increase in survival for every 10-point increase in global quality of life [91]. In patients with head and neck cancer, functional decline post-treatment was a strong predictor of mortality. Furthermore, Eldridge et al. demonstrated over three times the mortality rates in patients who declined to ‘Highly impaired’ when compared to those who maintained a stable functional status. Notably, the study used a unique measure of functional status specific to patients with head and neck cancer that focuses on impairment in diet, eating, and speech [92]. Morishima et al. reported similar findings across multiple cancer types, with increased mortality rates shown in patients with gastric (HR 1.39, 95% CI 1.09–1.77 for ‘Slight independence’ in ADLs; HR 3.34, 95% CI 2.81–3.97 for ‘Total independence’ in ADLs), colorectal (HR 1.64, 95% CI 1.24–2.17 for ‘Slight independence’; HR 2.86, 95% CI 2.43–3.36 for ‘Total independence’), and lung cancer (HR 1.24, 95% CI 0.96–1.59 for ‘Slight independence’; HR 3.21, 95% CI 2.80–3.68 for ‘Total independence’) [93]. In an analysis of patients with various solid and haematologic malignancies, functional decline in ADL (HR 2.34, 95% CI 1.75–3.12) but not IADL (HR 1.25, 95% CI 0.97–1.61) was prognostic for overall survival [57]. Notably, studies assessing the impact of functional decline and mortality are confounded by the fact that functional decline can occur due to progressive disease, and those with progressive disease have increased mortality independent of their functional status. While some studies account for cancer stage, these tend to focus on stage at diagnosis as opposed to stage as measured at the time of disease progression.

Performance evaluation measures, while not direct markers of functional status, appear to be particularly important prognostic factors. In older adults, both grip strength [94] and gait speed [95,96] are strong predictors of all-cause mortality. The evidence for these measures as predictors of mortality in patients with cancer is less conclusive. Celis-Morales et al. indicate that grip strength is associated with all-cancer mortality (HR 1.17, 95% CI 1.13–1.21), along with colorectal, lung, and breast cancers, but not prostate cancer, although the analysis did not adjust for cancer stage [94]. In patients with a pre-existing cancer diagnosis, there is some evidence supporting a link between poor grip strength and increased mortality [97,98,99], although Puts et al. did not report a statistically significant relationship between grip strength and survival, even after adjustment for stage of disease [100]. Notably, most of these studies suffer from small sample sizes, particularly when compared to the larger trials investigating grip strength in the general population. Additionally, adjustment for cancer stage is vital given that patients with advanced disease have higher mortality irrespective of their performance on measures of physical function. Less evidence exists linking gait speed to mortality in older patients with cancer, although Pamoukdjian et al. reported an association between slow gait speed and ‘early death’ in a cohort of older patients with various cancers [101].

The impact of functional status on the carers of older patients with cancer must also be acknowledged. There are various studies demonstrating that increased caregiver burden and psychological distress is associated with poor performance in ADLs in patients with various diseases [102,103]. There is similar evidence supporting this in patients with cancer, with most studies using the Zarit Burden Interview to assess levels of subjective caregiver burden. Wood et al. demonstrated a statistically significant link between declining ECOG score in patients with non-small cell lung cancer and worsening caregiver burden and activity impairment [104]. In older patients with advanced cancer, Semere et al. similarly found a link between patient functional status and higher caregiver burden [105]. Jansen et al. also reported an association between functional impairment and caregiver burden, while additionally reporting that over 16% of carers of older patients with cancer demonstrated high-to-severe carer burden at baseline [106]. Caregiver stress and burden play a clear role in increasing the likelihood of institutionalisation of older patients [107,108], further compounding the need to reduce carer burden associated with patient functional decline.

Given the use of functional and performance status measures in determining suitability for anti-cancer treatment, functional decline can also impact a patient’s ability to receive primary or subsequent lines of treatment. For example, ASCO recommends that patients do not receive cancer-directed therapy for solid tumours if their ECOG status is low (i.e., 3 or 4) [16]. In many older patients with cancer, this level of performance status is not uncommon, whether it be due to their cancer or other comorbidities, thereby precluding them from receiving treatment. Older patients with cancer with poor functional status pre-operatively are also at greater risk of post-operative complications and readmission, similarly reducing their suitability for surgery [109]. Although curative intent operations may not be likely in this cohort, patients may not be offered procedures with palliative intent due to their functional status, further accelerating the decline of their quality of life and function. It is also worth noting that patients may not be offered operations for comorbidities other than their cancer. Such operations, such as joint replacement surgery for osteoarthritis [110], can precipitate vast improvements in quality of life in older adults. The impact of cancer-related functional decline on the treatment of other disease is therefore another significant clinical implication to consider.

## 6. Prevention and Management of Functional Decline

Preventative measures and the management of functional decline largely overlap with those targeting frailty, given that frailty is effectively a marker for the potential for functional decline. While frailty cannot be directly treated, the components that comprise frailty can be targeted individually. Importantly, older patients can move between ‘frailty states’—that is, non-frailty, pre-frailty, and frailty. Gill et al. found that, while transitions from frailty to non-frailty were uncommon, up to 23% of a community-dwelling geriatric cohort were able to transition to lesser states of frailty [111]. Hurria et al. also suggest that early interventions aimed at improving functional status may play a role in a patient’s ability to ‘bounce back’ [56]. The principles of prevention and management of functional decline involve early detection, physical activity, and dietary interventions, both pre- and post-treatment. However, other factors that can precipitate or exacerbate functional decline, such as cognitive decline, falls, and polypharmacy, should also be targeted.

Early and regular measurement of functional status plays an important role in the management of cancer survivors. The CGA has been particularly lauded as a vital tool in assessing geriatric cancer patients due to its ability to predict treatment outcomes (and, therefore, trigger treatment modifications or cessation) and mortality, providing a more holistic picture of a patient’s health and wellbeing than simpler measures of functional status [18,112]. Kalsi et al. highlights this in a cohort of older chemotherapy patients, demonstrating that the CGA intervention reduced rates of adverse treatment-related outcomes and treatment modification while increasing treatment completion [113]. Given the association between poor functional status and symptom burden in older patients with cancer [114,115], it is logical that early detection of functional impairment is useful in triggering the implementation of interventions.

There are several national and international guidelines that support early geriatric assessment, including the National Comprehensive Cancer Network Guideline (NCCN) for Older Adult Oncology [116], the International Society of Geriatric Oncology (SIOG) Consensus on Geriatric Assessment in Older Patients with Cancer [28], and the ASCO Guideline for Geriatric Oncology (Practical Assessment and Management of Vulnerabilities in Older Patients Receiving Chemotherapy) [18]. While the latter has a focus on pre-treatment assessment in older adults with cancer, it nonetheless highlights the importance of the assessment of functional status [18]. The SIOG guideline suggests that a geriatric assessment should be performed in all patients ≥70, although other age cut-offs have also been used. The authors acknowledge the important role of the geriatric assessment for all patients, as opposed to just those receiving treatment, reporting that it can additionally act as a trigger for treating unidentified problems, and provide a better estimation of residual life expectancy. Notably, SIOG suggests that ECOG and KPS scores are a poor reflection of functional impairment in older patients with cancer, given that impairment can be present in spite of good performance status although no one tool has been found to be superior to another in the assessment of functional status [28]. Across all guidelines, there is debate as to which elements of the geriatric assessment are essential, and whether shorter tools can be used to determine whether a patient requires a more comprehensive assessment. However, it is clear that functional status should always be assessed in older patients with cancer, irrespective of treatment status.

Physical activity is essential in managing functional decline although many studies investigating exercise interventions use physical performance measures to assess outcomes as opposed to measures of functional status. A systematic review of 47 studies concluded there is ample evidence for its benefits in older adults. The findings suggest that exercise interventions can improve functional status, particularly if the intervention is performed multiple times per week, lasts 30–45 min for any one session, consists of multiple training types, and continues for at least 5 months [117]. Another meta-analysis of 34 randomised controlled trials of various cancer types (median sample size of 93) supports this, demonstrating a benefit for physical activity interventions including aerobic exercise and strength/resistance training on outcomes such as peak power output, 6MWT, and handgrip strength [118]. Although moderate-to-vigorous physical activity tends to be the focus of intervention for functional decline, lighter interventions may also have a benefit. Blair et al. demonstrated improvement in all SF-36 scores, Basic Lower Extremity Function, and Advanced Lower Extremity Function, following 1 year of either low–light and high–light intensity physical activity [119]. Arrietta et al. reports similar results using the Short Physical Performance Battery tool in a breast cancer survivor cohort. After 24 months, 29.8% of participants receiving standard care showed physical decline compared to 5.0% of participants in the physical activity intervention group (*p* < 0.01). Of note, the study found no significant improvements in other cancer types, nor after 12 months of the intervention, although this is likely due to the predominance of breast cancers in the study cohort and a small sample of other cancer types [120]. ‘Prehabilitation’, comprising pre-treatment exercise, nutritional intervention, and stress-reduction, may also have some benefit. An analysis of patients with colorectal cancer showed improvement in SF-36 scores and shorter 6MWT post-prehabilitation although this was only significant in patients with depressive symptoms pre-treatment [121]. Minnella replicated this in a cohort of patients prior to oesophagogastric surgery and demonstrated improvements in both pre- and post-operative functional capacity [122]. Prehabilitation is also undoubtedly beneficial for improving cardiorespiratory fitness [53,54], but further studies across various cancer types and treatment modalities are required to determine its impact on functional status.

Nutritional interventions have also shown promise in reducing functional decline. Ng et al. demonstrated the benefits of a vitamin and supplement package, along with the maintenance of a caloric surplus, on functional status. The intervention group of the randomised controlled trial (*n* = 151) showed decreases in frailty score at 3, 6, and 12 months, and were considerably more likely to experience frailty reduction compared to controls (OR 2.98, 95% CI 1.10–8.07) [123]. The previously described prehabilitation interventions both featured nutritional optimization [121,122], demonstrating some benefit for pre-treatment dietary intervention. There is also evidence to suggest vitamin D supplementation may be beneficial, particularly in vitamin D-deficient patients and those aged 65 years or older [124]. Oral supplementation of n-3 polyunsaturated fatty acids also appears to improve functional status outcomes. A double-blind, randomised, controlled trial of 40 patients with stage III non-small cell lung cancer demonstrated improvements in KPS score and physical activity multiple weeks post-supplementation [125].

Given the intimate link between functional decline and falls risk, it is crucial that falls prevention is also considered as part of functional status management [126]. It is also important to acknowledge the indirect complications of falls, particularly the potential for treatment interruption [127]. Sattar et al. report that the majority of oncologists do not routinely perform falls assessments, often due to a lack of time. Assessing falls risk as part of the standard oncological assessment may help provide early intervention to frail patients. If this is not feasible, clinicians should identify high-risk patients and flag them for referral [127]. Kagan et al. proposes an active approach to falls prevention through promotion of physical activity. The group further opines that supervised in-hospital exercise should be widely available in order to allow patients to maintain mobility [128]. This model is supported by Cameron et al. whose Cochrane review suggests exercise may be effective in preventing falls in hospitals. The evidence for patients in non-hospital settings, however, is less clear, and must be further investigated to ensure applicability to a predominantly outpatient-managed cancer-survivor cohort [129].

A similarly strong link exists between functional decline and polypharmacy in both older adults [130] and older adults with cancer [49]. This link is bi-directional, with functional impairment impacting a patient’s ability to manage and administer their own medications, but polypharmacy also has the ability to impair function through its impact on cognitive function, adverse drug reactions and toxicities, and falls. Nightingale et al. highlights the importance of pharmacist input on medication reconciliation and optimization [71]. Several validated tools such as the Beers criteria [131] and the STOPP/START criteria [132] can be used by both pharmacists and clinicians to identify patients at risk of polypharmacy. Tools created for use in older patients with cancer also exist and include the OncPal deprescribing guideline [133] and the PIP–CPC criteria [134] although both were developed for use in patients receiving palliative care. and the latter is a newer tool that has not yet been validated. There have also been several pilot studies investigating pharmacist-led assessments that can effectively reduce polypharmacy and medication-related adverse outcomes in older patients with cancer although randomised clinical trials investigating these interventions are required before they can be widely implemented into practice. Furthermore, clinician-led interventions that can be implemented at the bedside are also lacking and should be further researched [135,136,137,138].

## 7. Future Directions

There is ample evidence linking both cancer and cancer treatment to the onset of functional decline. There is a paucity, however, of time-to-event analyses in this area. Siddique et al. describes one of the few time-to-event analyses of functional decline in patients with cancer but provides analysis of grip strength and gait speed declines as opposed to true measures of functional status [139]. Future studies should aim to utilise longitudinal data with timepoints for functional status data in order to capture a more accurate picture of the acceleration of functional decline in older patients with cancer. Larger scale studies with patients suffering from various cancer types are also needed, given that many functional status studies in geriatric oncology are limited by small sample sizes although this may be difficult given that older patients with cancer who are functionally impaired may find it difficult to participate in research. One method by which this can be achieved is by including functional decline endpoints in oncology clinical trials, providing researchers with an avenue to conduct post-hoc analyses of functional status in trial participants.

Common tools used to assess functional status should also be validated in older patients with cancer. While these instruments may have validity in healthy or younger populations, older cohorts of patients with cancer are vastly different; in these patients, it may be difficult to accurately capture functional status. Head-to-head comparisons of functional status tools may also be warranted. Given the limitations of commonly used tools such as ECOG and KPS [28], quick, easy, and low-cost measures of functional status assessment should be investigated. It is also important that clinicians treating older patients with cancer conduct regular geriatric assessments, or at minimum, functional status assessments. Formal functional status assessments do not typically feature in routine cancer survivor care but should play a role in the long-term management of these patients to maximise quality of life and reduce mortality.

There are also several opportunities for research into interventions targeting the prevention and management of functional decline in cancer survivors. Multi-faceted prehabilitation and rehabilitation programs show promise but are not routinely used in clinical practice. Similarly, further investigation into nutritional supplementation and optimisation is required. Regarding falls prevention, much of the literature investigates patients in inpatient settings. Given that older cancer survivors are typically treated as outpatients, studies investigating preventative strategies in this setting are similarly necessary. Finally, randomised controlled trials investigating methods to reduce polypharmacy and, by extension, medication-related functional decline, in older patients with cancer are needed before any medication-optimisation tools can be implemented by pharmacists and clinicians.

Table 3 summarises some of the opportunities for future research and clinical practice in relation to cancer and functional decline.

## 8. Conclusions

Functional decline is an inevitable part of ageing, but it may be accelerated in older patients with cancer, due to both the tumour itself as well as anti-cancer treatment-related symptoms and toxicity. Functional status can be measured via various validated tools, some of which have been validated in cancer cohorts and are preferred for use in this population. The recognition of functional decline in older patients with cancer is particularly important, given that functional status is associated with quality of life, mortality, caregiver burden, and suitability for other medical treatment. Furthermore, functional decline can be both partly prevented and managed, especially if it is detected early. However, further research regarding the optimal measurement and management of functional decline in older cancer survivors is required before many of these tools can be routinely incorporated into clinical practice.

## Figures and Tables

**Figure 1 cancers-14-01368-f001:**
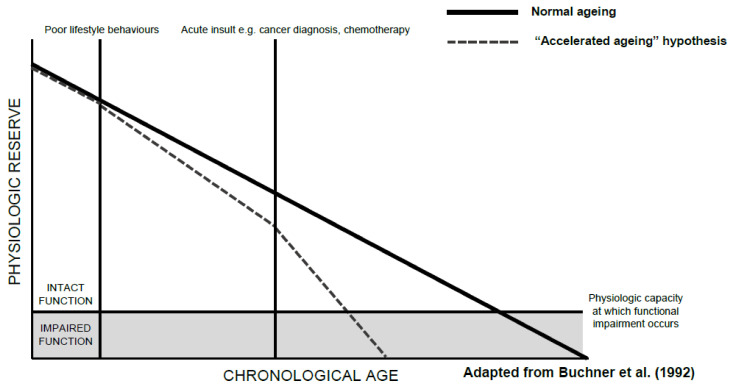
An “Accelerated Ageing” model of functional decline [11].

**Table 2 cancers-14-01368-t002:** Factors that may predict functional decline in older patients with cancer.

Patient Characteristics and Social Factors	Clinical Factors
Female sex [49] Older age [49] Unmarried [56] Poor financial status [70] Low educational attainment [10] Lack of health insurance [62]	Depression [55] Poor baseline functional status [55,57,62] Pre-treatment fatigue [56] Pre-treatment dyspnoea [56] Poor nutrition [57] Polypharmacy [57] Comorbidities [59] Cognitive impairment [62] Obesity [10]
**Cancer-related factors**	**Treatment-related factors**
Cancer type (e.g., breast, colorectal, lung) [71] Stage [71]	Chemotherapy [55,56,57] Radiotherapy [52] Surgical complications [63] Readmission after surgical hospitalisation [63]

**Table 3 cancers-14-01368-t003:** Future directions for research in relation to cancer and functional decline.

Opportunities for Research
Incidence of functional decline Use of time-to-event analysis in studies assessing the incidence of functional decline and its trajectory Recruitment of larger samples of patients with various types of cancer Inclusion of functional decline endpoints or functional status assessment in large-scale, oncology clinical trials Assessment Validation of functional status assessment tools in older patients with cancer Head-to-head comparisons of current functional status assessment tools Development of easy-to-use, cost-efficient, functional status assessment tools that can be quickly used by clinicians treating older patients with cancer Prevention and interventions Further investigation of the efficacy of interventions including physical activity and nutritional supplementation Falls prevention studies in the outpatient cancer-survivor setting Randomised clinical trials investigating the efficacy of deprescribing or medication-optimisation strategies
**Opportunities for Clinical Practice**
Regular functional status assessment of older patients with cancer +/− comprehensive geriatric assessment Implementation of evidence-based interventions (e.g., prehabilitation programs, vitamin supplementation, and deprescribing)

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
