# Peer review of "Functional Decline in the Cancer Patient: A Review"

_cancers, 2022, doi:10.3390/cancers14061368_

Round 1
Reviewer 1 Report
This is a very comprehensive review on the topic, particularly highlighting the focus for the vulnerable geriatric population.
The authors have described very well the concept of frailty, which is of importance here considering the discussion on functional decline, and summarized the tools used to measure frailty and functional status very well and clearly.
The review is very well-organized in terms of discussion on cancer-related/cancer treatment-related and other external factors which affects functional status in this scenario and all relevant references have been used to support the authors' analysis. The significance and implications of functional decline for the cancer patient is also discussed appropriately.
The authors have also clearly highlighted remaining knowledge gaps in this topic which would benefit from further study (the table was very useful).
Language is easy to understand throughout the review. The use of supporting figures and tables is appropriate.
Overall, a very comprehensive article which is satisfactory for publication.
Author Response
Thank you for your insightful comments and kind words regarding our manuscript. We appreciate the time taken to review our manuscript and provide feedback.
Reviewer 2 Report
Dear authors,
the chosen topic for the review is of importance, significance and of interest to the readers. However, in my opinion, the methodological approach is unclear and the quality of the presentation limited.
Following I will briefly explain some issues.
Research question: the research question „exploring current evidence" is aiming to cover a very broad area with the following 5 subtopics
- linking cancer and functional status
- outline assessments of functional status
- mechanisms driving functional decline
- strategies to prevent functional decline
- manage functional decline
in older patients with cancer
The chosen topic is of highest relevance in the oncologic community. The chosen subthemes however, are each so demanding, that it is very challenging to try to merge these in a single article with the necessary depth and scientific quality.
Method
It would be appropriate to define which kind of review the authors intended to do and to describe the methodology and respective reporting statement e.g. for scoping reviews e.g. http://www.prisma-statement.org/Extensions/ScopingReviews
In comparison to the listed aims, the search strategy provided in the supplement seems to be lacking a systematic approach to find relevant literature.
The authors report to have conducted an „initial scoping literature search“ with cancer; cardiovascular disease; stroke and then „a more comprehensive search“ with Functional decline OR functional status OR activity of daily living OR activities of daily living AND Cancer.
The authors report, that this search yielded a total of 195 articles.
Another search by the reviewer in PubMed Feb. 11 2022 with functional decline and cancer resulted in 1.199 and with the second search strategy in 2,779 articles.
In the main section, the authors provide an overview on assessments of functional decline. Firstly they focus on ADL, then on the concept of frailty followed by measures of performance status. Other domains like cognitive function are left out. They refer to „other tools that provide a comprehensive asseessment of older patients (p.115 line 103) and list the SF 36, which is generic QOL tool and the G8 whichis a screening instrument..
In the following parts the authors list several interesting studies however, the presentation lacks a systematic approach e.g. providing comparable information about design including the instruments used in the cited studies. Thus, regrettably the information value for the readers is limited.
Author Response
Thank you for these insightful comments and suggestions. We greatly appreciate the time taken to review our manuscript and provide feedback. We have detailed our responses to your suggestions below.
Research question: the research question „exploring current evidence" is aiming to cover a very broad area with the following 5 subtopics
- linking cancer and functional status
- outline assessments of functional status
- mechanisms driving functional decline
- strategies to prevent functional decline
- manage functional decline
in older patients with cancer
The chosen topic is of highest relevance in the oncologic community. The chosen subthemes however, are each so demanding, that it is very challenging to try to merge these in a single article with the necessary depth and scientific quality.
Author response: Thank you for the feedback. We do agree that this is a very broad topic with similarly broad subthemes. However, our manuscript represents a narrative review whose purpose is not to provide a systematic and comprehensive review of individual aspects of each theme, but instead provide a broad overview of the topics relevant to considerations of functional decline in older patients with cancer. The aim of the manuscript is therefore not to act as a definitive review of the totality of evidence that can be used to justify changes in clinical practice, but instead, to introduce the reader to the concept of functional decline in cancer and prompt further exploration of the topic through other publications cited in the article. We have now clarified this within the revised manuscript (line 48-54). We have also suggested to the Academic Editor that they consider retitling the manuscript to “Functional decline in the cancer patient: a narrative review” to ensure this is clear.
It would be appropriate to define which kind of review the authors intended to do and to describe the methodology and respective reporting statement e.g. for scoping reviews e.g. http://www.prisma-statement.org/Extensions/ScopingReviews
Author response: As mentioned above, this manuscript represents the findings and conclusions from a detailed and careful narrative review. Unlike that for systematic reviews and meta-analyses where the PRISMA statement is used, narrative reviews do not have a recognised structure to aid in their reporting.
In comparison to the listed aims, the search strategy provided in the supplement seems to be lacking a systematic approach to find relevant literature.
The authors report to have conducted an „initial scoping literature search“ with cancer; cardiovascular disease; stroke and then „a more comprehensive search“ with Functional decline OR functional status OR activity of daily living OR activities of daily living AND Cancer.
Author response: The terms used in our initial scoping search were “cancer; functional decline; activities of daily living”, not “cancer; cardiovascular disease; stroke”. These terms were mistakenly included in our supplementary material and did not form any part of the search strategy for this review. Our apologies for this error and we appreciate your bringing this to our attention. – we have rectified this in the updated version of our Supplementary Material.
The authors report, that this search yielded a total of 195 articles.
Another search by the reviewer in PubMed Feb. 11 2022 with functional decline and cancer resulted in 1.199 and with the second search strategy in 2,779 articles.
Author response: The difference between our search and the reviewers search is that ours was conducted on Ovid Medline (as mentioned originally in the Supplementary Material), while the reviewers search was conducted on PubMed. I have included a screenshot of the results of our Ovid Medline search below for reference.
While the two databases contain a near-identical number of articles, the method by which results are produced differs. PubMed typically returns more results by automatically expanding the search terms entered, therefore resulting in many results that are irrelevant to the search.1 Conversely, Ovid Medline provides results based on the precise search terms the user has entered. The greater accuracy and precision provided by Ovid Medline is why we used this database for our literature search. To illustrate this, I have included screenshots of searching for the term “cancer AND functional decline” on both PubMed and Ovid Medline; the former returns 21,185 results while Ovid Medline returns 316 results.
In the main section, the authors provide an overview on assessments of functional decline. Firstly they focus on ADL, then on the concept of frailty followed by measures of performance status. Other domains like cognitive function are left out. They refer to „other tools that provide a comprehensive assessment of older patients (p.115 line 103) and list the SF 36, which is generic QOL tool and the G8 which is a screening instrument.
Author response: Thank you for the feedback. In this section (“Section 2. Assessment of functional decline”) we define functional decline as a measured reduction in one’s ability to perform self-care, independent activities of daily living (ADL), and this is the main measure explored in the manuscript. While we then outline the concept of frailty and delineate its overlap with functional decline, this is to clarify the similarities and differences between functional status and frailty to avoid confusion; we do not focus on frailty as an outcome for the remainder of the paper. Similarly, we have included performance status measures as surrogate markers of functional status given that physical strength (e.g. grip strength) is often required to perform ADLs. However, we have acknowledged within the manuscript (line 90-95) that this correlation does not always hold, and that functional status can be maintained in the presence of poor physical performance. Regarding cognitive function, whilst we acknowledge that a reduction in ADLs can result from declines in cognitive function, our goal was to examine the downstream measure of functional status and hence, we have not explored this issue further in our manuscript.
Regarding the paragraph on comprehensive assessment tools, we have included this to introduce to the reader to the idea that functional status is only one part of the more holistic assessment that should be performed in older adults with cancer. However, we did initially and continue to acknowledge in line 99-104 of the revised manuscript that these tools have a broader scope than tools measuring functional status. As a result, we have not included studies in the later sections of this review that solely report results of comprehensive assessment tools, unless they have reported findings related to the functional status component of the tool. For example, the SF-36, a tool that was designed to measure health status and quality of life, features a functional status component that asks the patient to score their ability to perform various ADLs. Similarly, the G8 questionnaire, although a screening tool, does have a component (mobility) that provides insight into a patient’s ability to perform ADLs. However, we do acknowledge that the G8 tool provides a less comprehensive view of a patient’s overall health and have clarified this within our revised manuscript (line 114-115).
In the following parts the authors list several interesting studies however, the presentation lacks a systematic approach e.g. providing comparable information about design including the instruments used in the cited studies. Thus, regrettably the information value for the readers is limited.
Author response: Thank you for the feedback; we agree that the presentation of studies does not represent a systematic review. However, as per our previous comments, our paper is intended to be a narrative review that does not provide a systematic review of the literature, but instead provide a broader overview of functional decline in older adults with cancer. We have aimed to only include landmark studies and those with robust study designs, but we trust that the reader can use our manuscript as a starting point in their exploration of the topic.
References
- Ovid's Medline compared to PubMed [Internet]. Wolters Kluwer. 2021 [cited 28 February 2022]. Available from: https://wkhealth.force.com/ovidsupport/s/article/PubMed-vs-Ovid-s-Medline-1489081398582

Reviewer 3 Report
The paper gives a very nice overview concerning the problem of functional status in older individuals suffering from cancer. Several assessment tools are presented and discussed accordingly.
Furthermore, the authors give some advice for further research needed in this area of growing importance.
Author Response

(The authors gave the same response as above.)

Round 2
Reviewer 2 Report
Dear authors,
thank you for revising the manuscript and adressing the methodological issues in your point to point reply.
As introduction to the topic the article does provide a good overview despite the large scope.